# An Evaluation of Fengyun-3C Radio Occultation Atmospheric Profiles Over 2015–2018

**Jinde Wei [1], Ying Li [2], Kefei Zhang [1,3,*], Mi Liao [4], Weihua Bai [5], Congliang Liu [5], Yan Liu [6] and Xiaoming Wang [7]**

[1] School of Environment Science and Spatial Informatics, China University of Mining and Technology (CUMT), Xuzhou 221116, China; TS18160104P31@cumt.edu.cn

[2] State Key Laboratory of Geodesy and Earth's Dynamics, Innovation Academy for Precision Measurement Science and Technology (APM), Chinese Academy of Sciences (CAS), Wuhan 430071, China; liying@asch.whigg.ac.cn

[3] Satellite Positioning for Atmosphere, Climate, and Environment (SPACE) Research Centre, RMIT University, Melbourne, VIC 3041, Australia

[4] National Satellite Meteorological Centre, Chinese Meteorological Administration (NSMC/CMA), Beijing 100081, China; liaomi@cma.gov.cn

[5] Beijing Key Laboratory of Space Environment Exploration, National Space Science Center, Chinese Academy of Sciences (NSSC/CAS), Beijing 100019, China; baiweihua@nssc.ac.cn (W.B.); lcl@nssc.ac.cn (C.L.)

[6] National Meteorological Centre, Chinese Meteorological Administration (NMC/CMA), Beijing 100081, China; liuyan@cma.gov.cn

[7] Aerospace Information Research Institute, Chinese Academy of Sciences (AIRS/CAS), No.9 Dengzhuang South Road, Haidian District, Beijing 100094, China; wxm@aoe.ac.cn

\* Correspondence: profkzhang@cumt.edu.cn

**Abstract:** Fengyun-3C (FY-3C) is the first Chinese satellite that is capable of using the Radio Occultation (RO) technique to retrieve atmospheric profiles. This research evaluates the quality of FY-3C RO profiles including refractivity, temperature, and specific humidity by comparing with corresponding information from the European Centre for Medium-Range Weather Forecasts (ECMWF) Interim Reanalysis (ERA-Interim) data over the period of 2015–2018. The evaluation is carried out by calculating and analyzing mean systematic differences between FY-3C and ERA-Interim profiles and corresponding standard deviations over a selected spatial and temporal domain. Results show that the FY-3C RO profiles are overall with good agreements with the ERA-Interim data. Global mean refractivity systematic differences are within ±0.2% from 5 to 30 km altitude range with relative standard deviations of less than 2%. Global temperature mean systematic differences vary within ±0.2 K from a 10- to 20-km altitude range with standard deviations of less than 2 K. Global mean specific humidity differences are found to be within ±0.2 g/kg from 2 to 20 km with standard deviations of less than 1 g/kg. FY-3C profiles show visible latitudinal and altitudinal variations, while the seasonal variations are minor. Sampling errors of refractivity and temperature are also found to be larger at higher latitudinal regions due to RO events being less sampled in the polar region.

**Keywords:** fengyun-3C; GNSS; radio occultation; evaluation

## 1. Introduction

The Global Navigation Satellite Systems (GNSS) Radio Occultation (RO) technique is a robust limb-sounding technique to retrieve vertical profiles, such as refractivity, temperature, and specific humidity of the Earth's atmosphere [1,2]. The RO technique sets specially designed receivers onboard Low Earth Orbit (LEO) satellites to receive GNSS radio signals. As they propagate through the



atmosphere, these signals are bent due to the refractivity gradient. The accumulated bending angle can be retrieved by using GNSS observations and the orbits of both GNSS and LEO satellites based on the Geometric Optics (GO) and Wave Optics (WO) methods. The retrieved bending angles from two frequencies of the GNSS satellites are then linearly combined to remove first-order ionospheric errors [3]. They suffer from residual ionospheric errors and also other uncalibrated measurement errors, therefore a high-altitude initialization process is usually applied to the linearly combined bending angles to obtain "optimized" bending angles, which are then used to retrieve atmospheric refractivity by the Abel integration. By using refractivity, the atmospheric profiles of density, pressure, and temperature can be retrieved by employing the Smith-Weintraub, hydrostatic, and the ideal gas law equations in dry air conditions. In moist air conditions, profiles are retrieved by using prescribed temperature and humidity information using 1D-Variational (1DVar) or alternative approaches [4,5].

Compared with many other atmospheric detection methods, RO has several unique advantages. This includes high vertical resolution and accuracy in the upper and lower stratosphere (UTLS) as well as its global even distribution and self-calibration capability [6–11]. These advantages make RO data ideal for weather and climate studies. Thus far, the applicability of RO data has been investigated in many areas, including, but not limited to, numerical weather prediction [12–14], climate trend analysis [15] and weather regimes studies [16,17].

The first RO satellite to measure Earth's atmosphere, the Microlab-1 satellite from the GPS/Meteorology (GPS/MET) mission was launched in 1995 [18–20]. This pioneering mission provided about 150 occultation events per day. Evaluation results showed that the agreement of temperature profiles between GPS/MET and other observational techniques is mostly within 1 K from 1- to 40-km altitudes [19]. The success of the GPS/MET mission proves the capability of GNSS RO technique for observing the Earth's atmosphere. Following the proof of concept mission GPS/MET, RO receivers are considered to be one of the key "must have" payloads for observing the atmosphere in the design of many LEO satellite systems. For example, the Challenging Minisatellite Payload (CHAMP) satellite was launched by the US and Germany in 2000 [21,22]. It carried a BlackJack RO receiver developed by Jet Propulsion Laboratory (JPL), which was more powerful than the receiver on the MicroLab-1 satellite and sounded the atmosphere well in the lower troposphere region [23]. After CHAMP, the Gravity Recovery and Climate Experiment (GRACE) satellites were launched in 2002. In 2006, a constellation of six RO satellites (the Constellation Observing System for Meteorology Ionosphere and Climate—COSMIC) was launched by Taiwan and the US [24,25]. These six satellites provided about 2500 profiles per day. An open-loop tracking technique was used by the COSMIC, which enabled the system to receive RO signals in the lower troposphere [26,27]. After COSMIC, the Meteorological Operational Satellite Program (MetOp) series satellites were gradually launched. The MetOp satellites used a Global Navigation Satellite Receiver for the Atmospheric Sounding (GRAS) instrument to receive GPS signals [28]. Due to the robustness of the GRAS receivers and the orbital setup of the MetOp satellites, atmospheric profiles were found to be with high quality and accuracy [29,30]. Recently, the COSMIC-2 follow-on satellites were again launched in a group of six [31]. Its data are already available to the public. There are also many commercial RO satellites that are planned to be launched in recent years such as PlanetiQ and Spire missions [32–34].

The larger number of available RO satellites is expected to provide more observations of Earth's atmosphere. Hence, this is of great benefit for weather studies such as tropical cyclones and sudden stratospheric warming events [35–37]. Aiming to provide better weather and climate services, the Chinese Meteorology Agency (CMA) launched the Fengyun-3C (FY-3C) satellite which is the first Chinese satellite that carries a RO receiver to provide vertical profiles of the Earth's atmosphere [38]. It is a sun-synchronized polar-orbiting satellite with an altitude of 836 km. The RO receiver onboard, named Global Navigation satellite system Occultation Sounder (GNOS), was designed and developed by the National Space Science Centre of the Chinese Academy of Sciences (NSSC/CAS) [39,40]. It is capable of receiving signals from both GPS and BeiDou navigation satellite System (BDS) constellations. The GNOS consists of three antennas: three Radio Frequency Units

(RFUs), an Electronic Unit (EU), which uses highly dynamic, sensitive signal acquisition and tracking techniques. The three RFUs are installed close to their corresponding antennas by using sharp cavity filters, to reduce the loss of the cable between antennas and RFUs. The EU is the major component of GNOS, which accomplishes the GNSS remote-sensing signals' acquisition and tracking as well as the real-time positioning and carrier phase observations. In FY-3C GNOS design, the different features of BDS and GPS signals have been taken into account, and it can observe both the neutral atmosphere and ionosphere by using BDS and GPS signals. In the lower troposphere, GNOS employs the open-loop mode, which allows an accurate tracking of RO signals in areas where multipath often occur. Further details on GNOS receivers can be found in Sun et al. [38] and Bai et al. [39]. The FY-3C data processing strategy has been introduced in several papers. For instance, Li et al. [41] and Bai et al. [39] introduced the orbit determination and the excess phase calculation of the FY-3C system, respectively. Sun et al. [38] introduced the level-1 and -2 data processing chain of the FY-3C data. Initial evaluation results showed that the retrieved atmospheric profiles are of comparable quality to those of the other RO missions [42–44]. Liao et al. [42] used several months' RO data to evaluate precision against European Centre for Medium-Range Weather Forecasts (ECMWF) Interim Reanalysis (ERA-Interim) data. The refractivity mean systematic differences are found to be within ±0.2% from a 5- to 25-km altitude with the overall standard deviations amount of 1%. Liao et al. [43] evaluated the performance of FY-3C against radiosonde (RS) data by using several months of FY-3C observations. It is found that the standard deviation of the GNOS refractivity is about 1.85% in the vertical altitude range below 25 km, is greater than 2% of 0-5 km and there is a weak negative average deviation. Xu et al. [44] used Integrated Global Radiosonde Archive 2 (IGRA2) radiosonde data to evaluate the performance of FY-3C profiles. Refractivity show small mean differences against radiosondes with standard deviations less than 5%. Small negative differences are found in temperature and standard deviations of specific humidity are less than 1.2 g/kg below 5 km.

Though FY-3C profiles have been evaluated in the above studies, they have not yet been evaluated using long-term ERA-Interim data. The latitudinal and seasonal variations of refractivity, temperature and specific humidity profiles have not been fully investigated. Furthermore, there are no investigations over the sampling error of FY-3C profiles. This paper first evaluates FY-3C atmospheric profiles including refractivity, temperature, and specific humidity profiles against ERA-Interim data over the period of 2015–2018. Then the temporal and spatial variations of FY-3C atmospheric profiles and sampling errors are analyzed. Section 2 introduces the data and methodology of the study. The evaluation results are shown in Section 3. Section 4 discusses the results in Section 3 and a conclusion is given in Section 5.

## 2. Data and Methodology

### 2.1. FY-3C RO Data

FY-3C data processing consists of three main parts. The first part is the precise orbit determination (POD) of FY-3C satellites using the precise orbit and clock products from the International GNSS Service (IGS). Secondly, excess phase data as a function of time are calculated using the orbits of GNSS and FY-3C satellites as well as the downloaded occultation observations. Thirdly, the Radio Occultation Processing Package (ROPP) version 6.0 developed by the Radio Occultation Meteorology Satellite Application Facility (ROM-SAF) is used to retrieve atmospheric profiles based on the obtained excess phase data [38] In retrieving atmospheric profiles, bending angle profiles are retrieved using the Geometric Optics (GO) method above 25 km, while below the Wave Optics (WO) method is used for calculating bending angles [45–47]. In order to minimize ionospheric error and other measurement noises, the approach developed by Gorbunov [48], which combining the ionospheric correction and high altitude initialization together, is applied on bending angles of GNSS two frequencies to provide optimal bending angle profiles. In the data processing of FY-3C, the Mass-Spectrometer-Incoherent-Scatter (MSIS) model is used to provide background bending angle profiles for the noise reduction [49].

Based on the optimized bending angle profiles, refractivity profiles are then retrieved through the Abel transform. Based on refractivity, temperature, specific humidity and other moist profiles are retrieved using a 1D Variational (1DVar) process [50]. The Chinese Global Forecast Model (CGFM), T639L60 is used to provide the background profiles for 1DVar. Details of the FY-3C data processing can be found in Sun et al. [38].

FY-3C mission provides three RO products, i.e., Atmospheric Refractive Profiles (ARP), Atmospheric Temperature Profiles (ATP), and Atmospheric Moisture Profiles (AMP). The ARP products consist of atmospheric refractivity profiles and its related data such as bending angles and impact heights as well as GNSS to LEO azimuth information which is used to obtain atmospheric profiles. The ATP products provide temperature and pressure profiles and the AMP products provide specific humidity and geopotential profiles. Both temperature and specific humidity profiles provided by ATP and AMP are obtained using the 1DVar module of ROPP with some updates.

In this paper, four years of FY-3C RO data from 2015 to 2018 are used to evaluate the quality of refractivity, temperature, and specific humidity profiles. Figure 1 shows the numbers of monthly available products from January 2015 to December 2018. The numbers of products in most months are above 12,000, while it is as low as 8000 for some months. Data were missing for three months from June to August 2015. The numbers of ARP products are larger than that of ATP and AMP products, while the numbers of ATP and AMP are almost the same. The differences among the numbers of the three products are caused by quality control schemes implemented at each processing stage. If a retrieved profile cannot pass the quality control check, it will be rejected from the calculation of the next step.

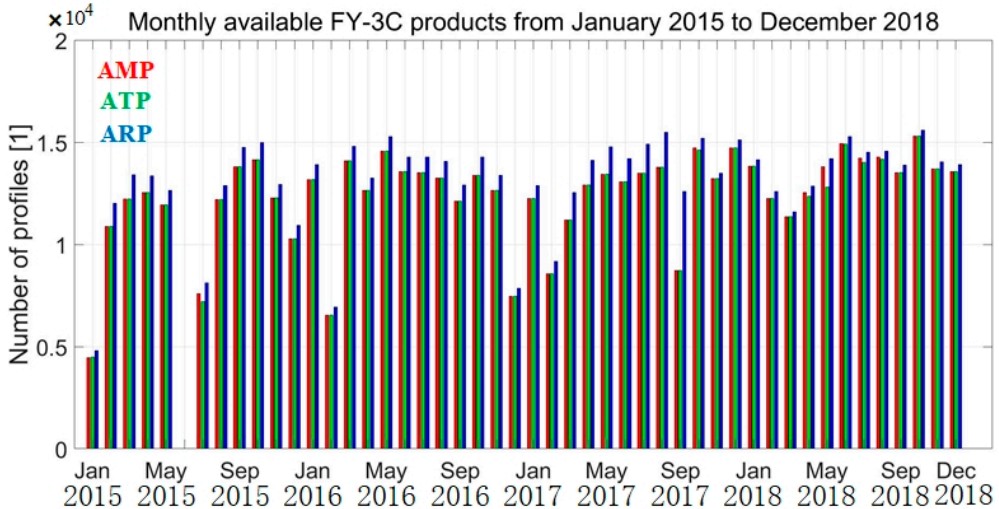

**Figure 1.** Monthly available FY-3C AMP, ATP and ARP products from January 2015 to December 2018.

Figure 2 shows the distribution of FY-3C RO events of an exemplary day of January 15, 2017 (upper panel), the numbers of daily available FY-3C RO products in January 2017 (middle panel) and numbers of FY-3C ARP products for January 2017 in 10° latitudinal intervals (bottom panel). The marked three stars are the three selected events used for single RO events illustration performance in Figure 3. The middle panel shows that the daily RO events vary from 600 to 1000 for most days.

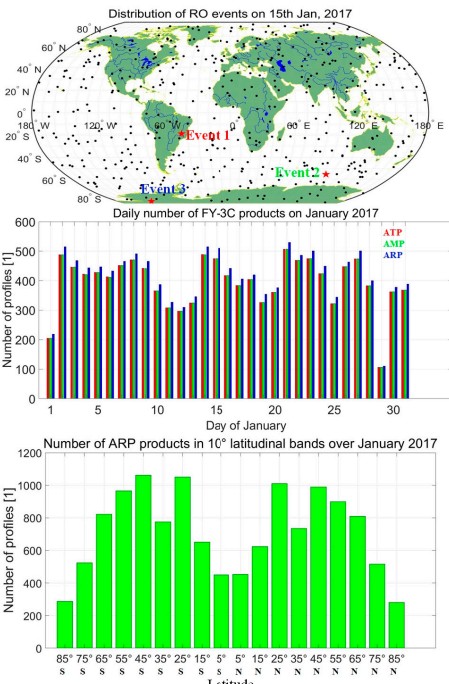

**Figure 2.** Global distribution of RO events on 15 January 2017 (top panel), number of daily products for January 2017 (middle panel) and numbers of FY-3C ARP products for January 2017 in 10° latitudinal intervals (bottom panel); the marked three stars are the three RO events for single RO performance illustrated in Figure 3.

### 2.2. ERA-Interim Data

ERA-Interim is a global atmospheric reanalysis that is available from 1 January 1979 to 31 August 2019. It provides atmospheric observations every day at 00:00, 06:00, 12:00, and 18:00 (UTC). In this study, we use ERA-Interim's daily data at the standard pressure level of 37 layers. Considering the horizontal footprint of RO is about 300 km, the 2.5° × 2.5° grid is selected for the evaluation. The ERA-Interim data provide temperature and specific humidity data on geopotential height and they are converted to be based on MSL altitude for our comparison to RO below. Refractivity $N$, which is not provided by ERA directly, is estimated by the Smith-Weintraub equation.

In order to obtain collocated profiles, we do a spatial and temporal interpolation according to the time and location of each coming in RO event. We first interpolate in time to find the four profiles of the same time of the RO event at the four grids that surrounding the RO location. We then bilinearly interpolate the four surrounding profiles to the RO location. Finally, we use a spline interpolation method to vertically interpolate the collocated profiles on our standard vertical grids from 0.2-km above the surface to 40.0 km with 0.2 km as a step.

### 2.3. Methodology and Quality Control

The first step of our method is to calculate the difference profiles of refractivity, temperature and specific humidity by subtracting RO profiles with collocated ERA-Interim profiles. Refractivity differences are calculated as relative values by dividing by the ERA-Interim data. We first examine the performance of individual RO profiles by selecting three exemplary RO events locating in different latitudinal bands. We then calculate global mean systematic differences and corresponding deviations of each year from 2015 to 2018 to examine the yearly characteristics of FY-3C profiles. After that, the seasonal variations are analyzed by calculating differences over four seasons of December, January and February (DJF), March, April and May (MAM), June, July and August (JJA) and September, October and November (SON). Considering the seasonal differences of northern and southern hemispheres, we show the results for both hemispheres. Latitudinal variations are investigated by calculating mean

systematic differences and standard deviations of RO against ERA-Interim profiles in five latitudinal bands of 60° N–90° N, 20° N–60° N, 20° S–20° N, 20° S–60° S, 60° S–90° S. Finally, sampling errors, which are used to evaluate whether FY-3C observations are evenly sampled in temporal and spatial domain, are investigated.

In order to make sure that high quality profiles are used for our comparison, we apply a series of quality control system. First of all, RO events with bad quality indication marked in the FY-3C satellite data processing system files are extracted. Secondly, RO events with refractivity deviation higher than 10% for the altitude of 5 to 25 km and temperature deviation higher than 20 K for the altitude of 8 to 25 km are excluded. We reject RO events if any of the criteria is not satisfied. After applied the above procedures, about 15% of the RO events were removed.

## 3. Results

### 3.1. Individual Profile Comparison

Figure 3 shows individual RO refractivity, temperature, and specific humidity profiles (left panels) and their corresponding difference profiles (right panels) of three exemplary RO events located in the tropical, middle latitudinal and polar regions of the southern hemisphere (see the red stars in the top panel of Figure 2). RO refractivity differences of all the three events are generally similar with differences varying within ±2% for most altitude levels from 10 to 40 km. Below 10 km, refractivity differences increase to be more than 4%. Temperature differences of all three events vary within ±2 K from 10 to 30 km. Above 30 and below 10 km, temperature differences are larger with values varying within ±4 K, and differences of Event 3 which locating in polar regions are largest among the three events. Specific humidity differences of Event 1 are largest due to its location in the tropics. The specific humidity difference of the three events are similar.

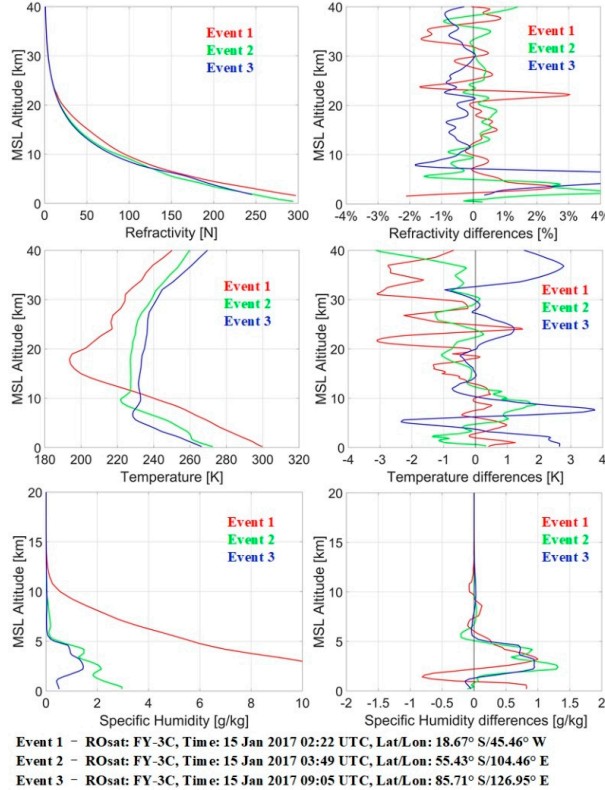

Event 1 – ROsat: FY-3C, Time: 15 Jan 2017 02:22 UTC, Lat/Lon: 18.67° S/45.46° W
Event 2 – ROsat: FY-3C, Time: 15 Jan 2017 03:49 UTC, Lat/Lon: 55.43° S/104.46° E
Event 3 – ROsat: FY-3C, Time: 15 Jan 2017 09:05 UTC, Lat/Lon: 85.71° S/126.95° E

**Figure 3.** FY-3C refractivity, temperature, and specific humidity profiles (left panels) and their corresponding differences against collocated ERA-Interim profiles of three exemplary RO events as indicated in Figure 2.

### 3.2. Global Statistics

Figure 4 shows the global mean systematic differences and corresponding standard deviations of RO refractivity, temperature, and specific humidity on a yearly basis for the 2015–2018 period. The number of RO events of each year are also indicated in the figure panels. It can be seen from the figure that the yearly statistical differences are gradually reduced with time. The differences in 2015 are found to be larger. With the evolution of time, the number of RO events is increased and the performance of the FY-3C profiles are better due to a further upgrade of retrieval algorithms and quality control issues.

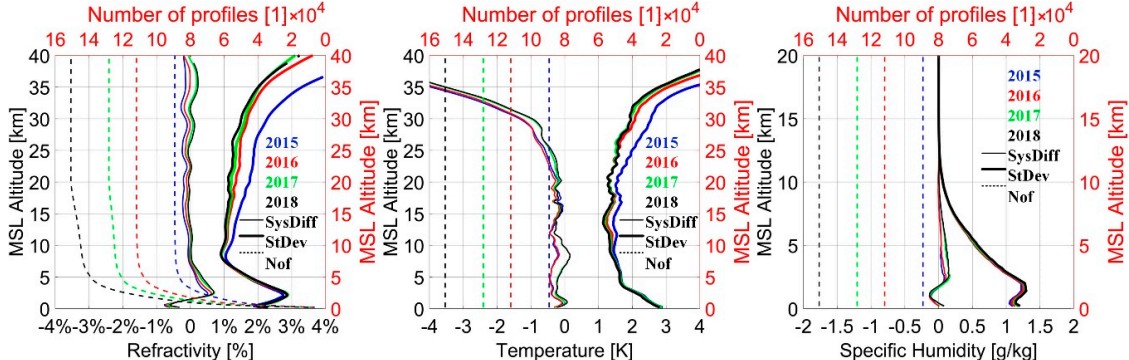

**Figure 4.** Global mean systematic differences (SysDiff, thin lines) and corresponding Standard deviation (StDev, thick lines) of FY-3C refractivity, temperature, and specific humidity profiles for the years from 2015 to 2018. Dashed lines indicate the number of profiles (Nof) used for the yearly statistics.

Refractivity systematic differences of all years are varying within ±0.2% from 5 to 30 km with standard deviations of less than 2%. Below 5 km, refractivity errors increase due to several reasons, such as the existence of water vapor and subsequent multipath, larger horizontal gradients, etc. From 2 to 5 km, there are systematic positive biases exceeding 0.5% are found with standard deviations exceeding 2%. Further efforts will be carried out to investigate the origins of these positive biases. Below 2 km, negative refractivity biases of about -0.8% are found. These negative biases are caused by super-refraction, which occurs when the vertical refractivity gradient exceeds the critical value ($dN/dz < -157$ N-units km$^{-1}$). It often results from the combination of a sharp vertical gradient in moisture and a temperature inversion frequently observed at the top of the marine atmospheric boundary layer [51]. The values of such negative biases are typically around −0.5% to −1% over large regions, with values exceeding −5% in regions where ducting is prevalent, such as off the west coast of the U.S. of South America [6,51–55]. Standard deviations of refractivity are about 3% at 2 km and then decrease to 2% close to the surface.

In Figure 4, temperature mean systematic differences of 2017 and 2018 are smaller than that of 2015 and 2016 due to an improved performance of retrieval algorithms of the FY-3C system. From 0 to 10 km, the systematic differences of 2017 and 2018 are within ±0.2 K and that of 2015 and 2016 are about than −0.5 K. From 10 to 25 km, temperature systematic differences show slight consistent negative values varying from −0.1 to −0.6 K. These negative differences are consistent with the findings in Xu et al. (2019) where they compared FY-3C temperature profiles with RS profiles. Above 30 km, systematic differences of temperature increase rapidly and exceed −4 K above 35 km. Temperature standard deviations of all years are generally within 2 K in the altitude range of 5 km to 30 km. Above a 30-km altitude, temperature standard deviations increase rapidly and also exceed 4 K above 35 km. The number of profiles of temperature and specific humidity are the same till the surface, while number of refractivity profiles decrease significantly below 5 km. This is because temperature and specific humidity are obtained using the 1DVar process, which combine RO observed information with prescribed background information. In the 1DVar process of FY-3C, if observed information were missing, background profiles would then be used to provide the optimal profiles directly. Since

background profiles are always available till the surface, so the retrieved temperature and specific humidity profiles are also always available till the surface. Due to the increased dependence on the background values, the differences of temperature and specific humidity from ERA-Interim are more closely related to the background values rather than RO below 2 km in this study.

Specific humidity systematic differences are generally within ±0.2 g/kg over all altitudinal levels. Below 3 km altitude, the differences are negative with the highest amount of −0.2 g/kg. Specific humidity standard deviations are more than 1 g/kg at the surface and gradually decrease to less than 0.1 g/kg at 10 km.

### 3.3. Seasonal Variations

Figure 5 shows the global statistical errors among different seasons of December, January, and February (DJF); March, April, and May (MAM); June, July, and August (JJA); and September, October, and November (SON) for both northern (upper panels) and southern hemispheres (bottom panels). The month of December is from the year of 2016; all other months are from 2017. The numbers of RO events in different seasons are also provided in Figure 5. It can be seen from the figure that the numbers of events are largest in JJA season and are smallest in DJF season. This is due to reason that the numbers of original products of JJA season are largest and the numbers of DJF are smallest (c.f. Figure 1). All difference and standard deviations profiles show overall small seasonal variations. Above 30 km, both refractivity and temperature of summer seasons (i.e., JJA for northern hemisphere and DJF for southern hemisphere) are always smaller than that of the winter seasons (i.e., DJF for northern hemisphere and JJA for southern hemisphere). The main reason is the differences between estimated observed and background bending angle uncertainties, which then determine the weights of observed and background bending angle used in the optimized bending angle. If the optimized bending angles use more weight from observed/background profiles, they agree more with ERA-Interim bending profile. The estimated uncertainties in winter and summer seasons are different, therefore the resulting optimized bending angles show different characteristics in different seasons and those of the winter season show less agreement with ERA-Interim. These differences in bending angles propagate into refractivity and temperature profiles in atmospheric retrievals and we therefore also see the seasonal variations in refractivity and temperature. In future, more efforts will be carried out to minimize these seasonal variations above 30 km. Finally, specific humidity errors also show seasonal variations with largest errors found in summer seasons where larger amount of water vapor exist.

### 3.4. Latitudinal Variation

Figure 6 shows the statistical errors of refractivity, temperature, and specific humidity in five latitudinal bands of 60° N–90° N, 20° N–60° N, 20° S–20° N, 20° S–60° S, 60° S–90° S. In middle latitudinal bands, i.e., 20° N–60° N and 20° S–60° S, the mean difference varies within ±0.2% above 5 km. Below 5 km, the differences increase and also show positive biases of about 0.7% from 2 to 5 km which are similar to the global statistics shown in Figure 4. Negative biases reaching about −1% are found at lowest few altitude levels. These negative biases are of similar magnitudes found by many other studies [42]. Standard deviations of these two latitudinal regions vary from about 1% at 10 km to about 3% at 40 km. Below 10 km, standard deviations increase to about 3% at 2 km and then decrease to 2% near the surface. In tropical regions, mean differences and standard deviations above 10 km are similar to that of middle latitudinal bands. However, below 10 km, strong positive differences are found from 3 to 10 km with values exceeding 2% with standard deviations exceeding 4%. These strong positive differences are consistent with the findings found by Liao et al. [42], but with slightly larger magnitudes. Below 2 km, the negative biases are smaller than that found by Liao et al. [42]. We cannot determine the exact causes of these small negative biases in tropical regions for now, further efforts are required. In the latitudinal band of 60° N–90° N, refractivity systematic differences show consistent negative values varying from −0.2% to −0.3%, while in 60° S–90° S, refractivity systematic differences show consistent positive values varying from 0.2% to 0.3%.

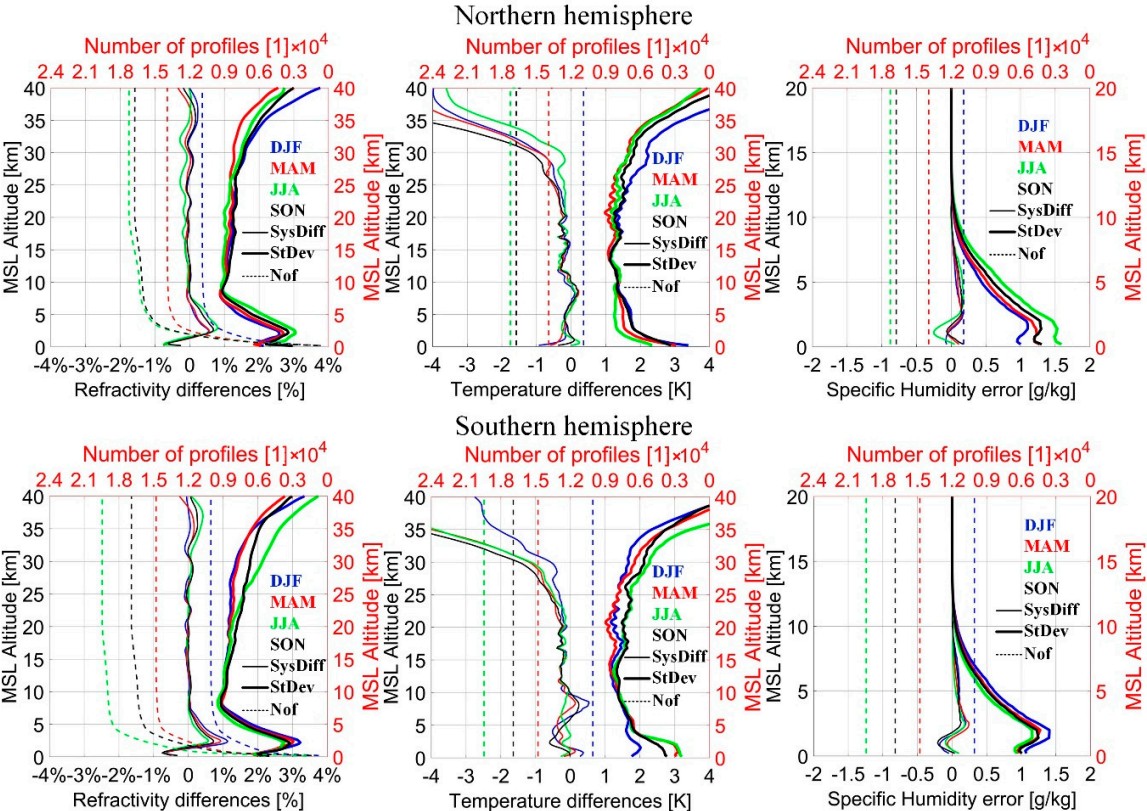

**Figure 5.** Systematic differences (SysDiff, thin lines) and standard deviations (StDev, thick lines) for the seasons of December, January, and February (DJF); March, April, and May (MAM); June, July, and August (JJA); and September, October, and November (SON); the December profiles are from 2016, all other months are from 2017; Dashed lines indicate the number of profiles (Nof) used for the seasonal statistics.

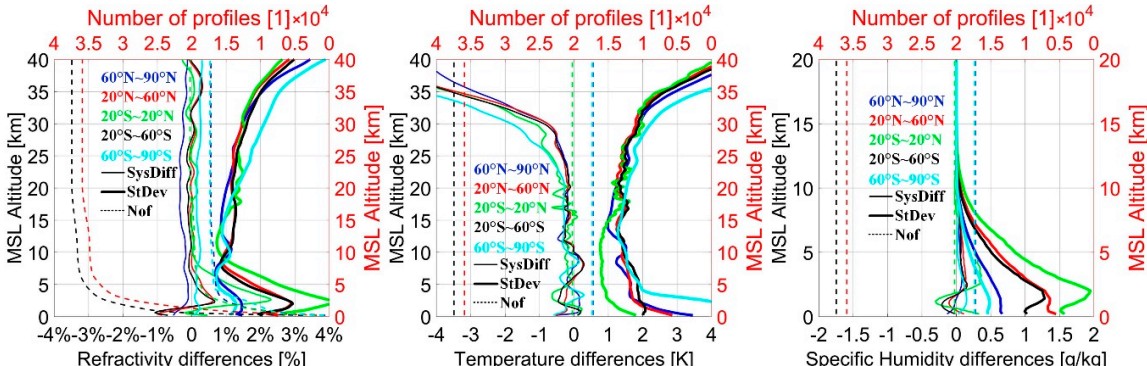

**Figure 6.** Systematic differences and standard deviations of FY-3C refractivity, temperature, and specific humidity profiles for the latitudinal bands of 60° N–90° N, 20° N–60° N, 20° S–20° N, 20° S–60° S, 60° S–90° S. Statistics are obtained by considering the FY-3C profiles in 2017. Dashed lines indicate the number of profiles (Nof) used for the corresponding latitudinal band.

There could be several reasons for these systematic positive and negative biases in polar regions. First of all, improper weighting of observed bending angles in the ionospheric correction and noise reduction processes. Since FY-3C data processing system uses ROPP for processing, and the empirical error estimation functions in polar regions may not suitable for FY-3C bending angles. Therefore, the resulting optimized bending angle are biases. These systematic biases in bending angle were then propagated into refractivity profiles. Secondly, the background profiles used by FY-3C have systematic

discrepancies against ECMWF in polar region. Further efforts are required to investigate the origins of the refractivity biases in polar region.

The systematic differences of temperature in the polar region are also larger than those in subtropical and tropical regions, within ± 0.5 K in all latitudinal bands for the altitude up to 25 km while increase rapidly and exceed −4 K above the 25 km altitude range. Temperature standard deviations are generally within 2 K within the 5–30 km altitude and the values increase rapidly and reach 4 K above 30 km in altitude. At the lower altitude levels, standard deviations of the tropical region are relatively smaller. This is because the 1DVar process uses more information from background which agree more with ERA-Interim profiles in tropical region. The systematic differences of specific humidity show clear seasonal variations with largest values found in summer season and smaller errors found in winter season.

Figures 7 and 8 further illustrate latitudinal and seasonal variations of the differences of FY-3C refractivity, temperature and specific humidity of monthly statistical errors in 10° latitudinal intervals for January and July of 2017. Looking at Figure 7, which shows results of systematic differences, refractivity differences show positive values varying from 0.2% to more than 2% over the height range from 2 to 5 km in tropical and middle latitudinal regions. These positive biases centered on tropical regions and decrease with increasing latitude. Negative refractivity biases are found below 2 km, but with only a thin layer, especially in tropical regions. In the polar region of the southern hemisphere, positive biases (especially for July) varying around 0.2% are found, while in the polar region of the northern hemisphere, negative biases varying around to −0.4% are found. These biases found in polar region are consistent to the findings in Figure 6. Looking at temperature differences shown in middle row, the differences are with small negative values for most cases indicating that FY-3C retrieved temperature are generally cooler than that of ERA-Interim. Specific humidity differences are largest from 2 to 8 km over tropical and middle latitudinal regions with largest values center over tropical regions. Differences decreases with increasing to higher latitudinal regions. This indicates that FY-3C moisture products over this height range are generally more moisture than that of the ERA-Interim results. Below 2 km, negative differences up to -0.2 g/kg are found indicating FY-3C humidity measurement over this height range are generally drier than ERA-Interim.

Figure 8 shows the corresponding standard deviation profiles in 10° latitudinal intervals for January and July of 2017. It can be seen from this figure that refractivity standard deviations have clear vertical variations. From 10 to 30 km, the values are varying from 1% to 2%. Below 10 km, the values increase to more than 2% below 2 km. Refractivity standard deviations also show small seasonal variations above 20 km. In January, 2017, refractivity standard deviations are larger in the polar region of northern hemisphere, while in July, the values in the polar region of southern hemisphere are larger. The temperature standard deviations of July show also clear vertical, latitudinal and seasonal variations. From 10 to 20 km, values are varying around 1 K, which are smaller than that above 20 and below 10 km. Above 20 km, temperature standard deviations increase with height and are largest in the winter season of both hemispheres. Below 10 km, values in tropical regions are larger than that in middle and high latitudinal regions. Specific humidity standard deviations are found to be largest in the tropical region below 4 km where a larger amount of water vapor exists.

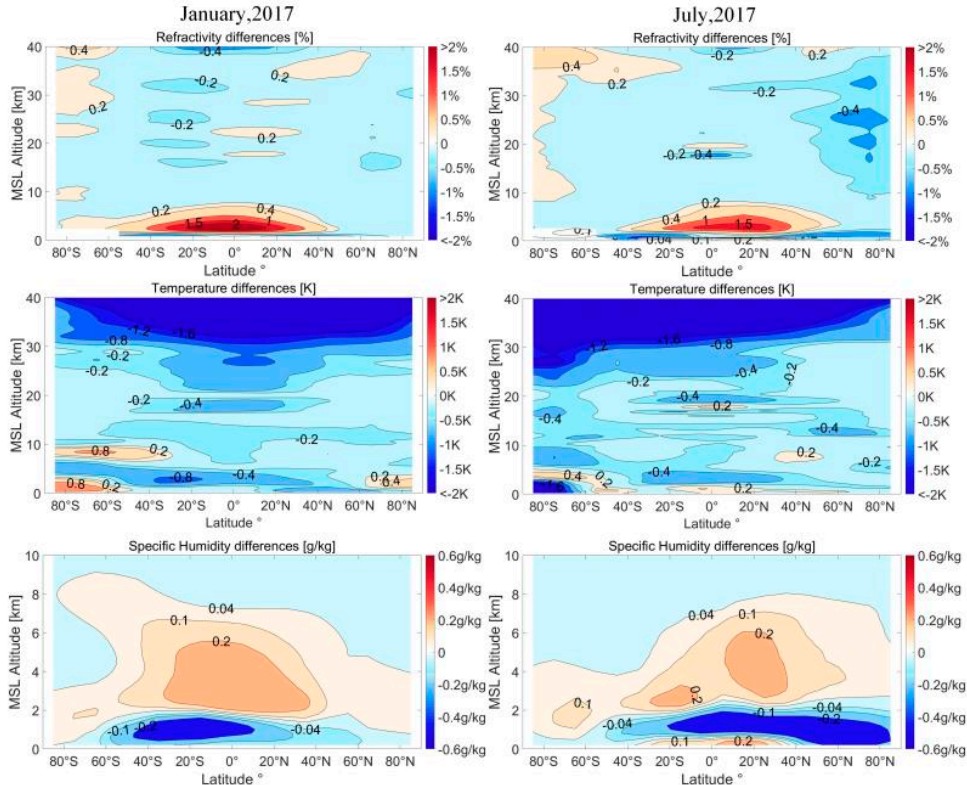

**Figure 7.** Latitudinal and altitudinal variations of systematic differences in refractivity (top panels), temperature (middle), and specific humidity (bottom) parameters in January and July of 2017.

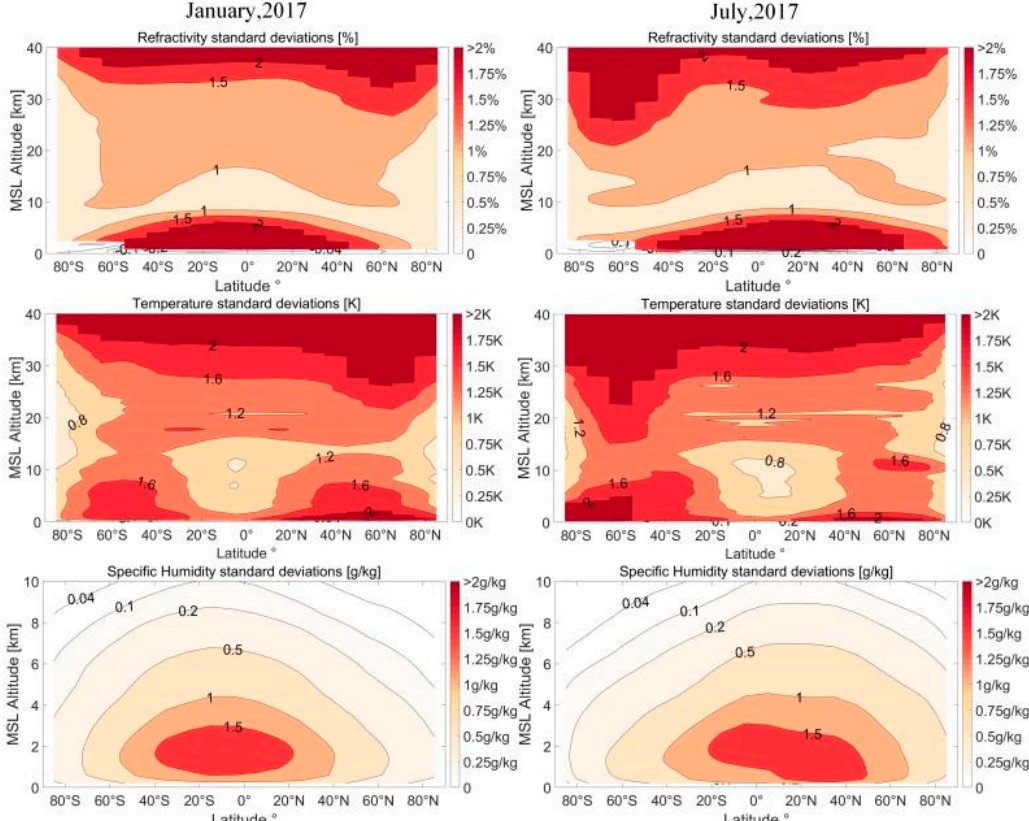

**Figure 8.** Latitudinal and altitudinal variations of standard deviations in refractivity (top panels), temperature (middle), and specific humidity (bottom) parameters in January and July of 2017.

### 3.5. Sampling Errors

A sampling error is caused by discrete sampling times and locations of RO measurements [7,56–59]. It consists of both random and systematic components. The random component is caused by the atmospheric variability not adequately sampled by RO events. The climatic mean is affected by a sampling error if observations miss some parts of the atmospheric variability. Even though some low atmospheric variability might be captured by a smaller number of measurements, the true state of the atmosphere requires more and evenly distributed RO observations in the selected temporal and spatial domain. The systematic component of sampling error is caused by systematic spatial and temporal under sampling of the atmospheric variability, e.g., due to RO events never sampled in certain modes of variability. Sampling error is one of the main error sources when constructing RO climatology for climate and weather studies. The sampling error could be larger if there are some observational gaps in the selected spatial and temporal domain.

RO sampling error is usually calculated by considering the differences between the mean of all collocated reference profiles of a bin $\bar{x}_{\text{coloc}}$ and a fully sampled mean profile $\bar{x}_{\text{full}}$ over a fully and regularly sampled mean profile. Then, the sampling error $s_{\text{samplErr}}$ for a zonal mean monthly mean field $(z_i, \varphi_j, t_k)$ can be calculated as [57]:

$$s_{\text{sampleErr}}\left(z_i, \varphi_j, t_k\right) = \bar{x}_{\text{coloc}}\left(z_i, \varphi_j, t_k\right) - \bar{x}_{\text{full}}\left(z_i, \varphi_j, t_k\right) \tag{1}$$

where $z_i$ is the altitude, $\varphi_j$ is the latitudinal bin (10° intervals in this study), and $\varphi_j$ is the temporal period which is month in this study. This sampling error is based on the ERA-Interim data.

Refractivity and temperature sampling errors for January and July 2017 are presented in Figure 9. Sampling errors of specific humidity are similar to temperature errors and are therefore not shown. The figure indicates clear variations with respect to altitude, latitude, and season for both parameters. For each parameter, relatively higher sampling error amounts are obtained at the higher altitude around the polar region. Refractivity sampling errors for January 2017 are highest above the 30-km altitude and beyond 60° N with values exceeding −1.5%. In July 2017, error amounts for the same parameter are relatively highest from 20 to 40 km and exceed 0.5% beyond 60° S. Temperature sampling errors show similar characteristics as that of refractivity. Errors of January 2017 are largest above 20 km altitude beyond 60° N with values exceeding −1.8 K while it is largest above 30 km and beyond 60° S with values reaching −1.2 K for July 2017.

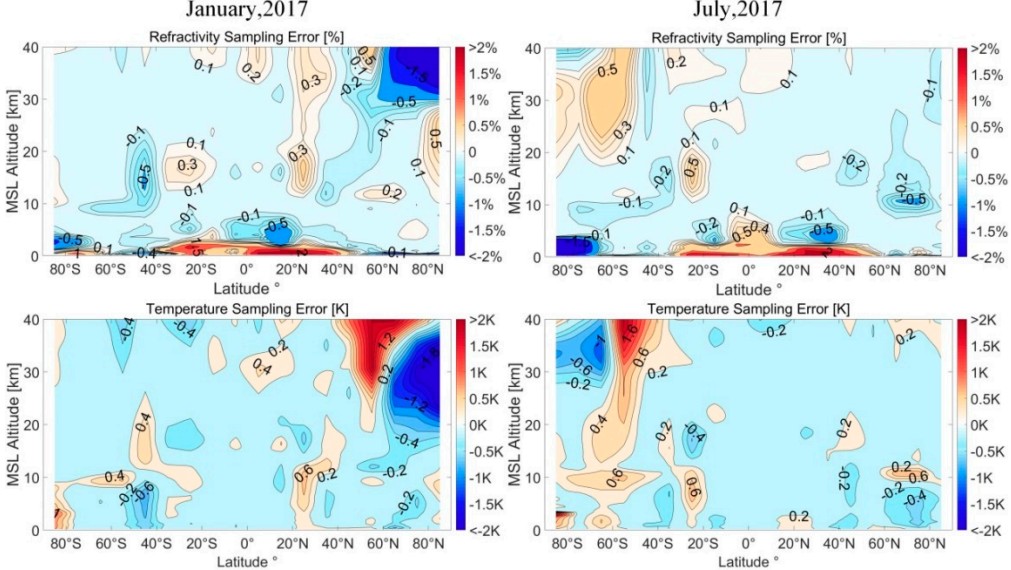

**Figure 9.** Refractivity and temperature sampling errors for January (left panels) and July (right panels) of 2017 with 10° latitudinal interval bands.

Figure 10 shows the monthly variation of FY-3C RO events in every 10° latitudinal band and the sampling errors from 20 to 25 km in 2017. The top panel in the figure indicates that the numbers of FY-3C RO events in polar and tropical regions are relatively smaller than those in middle latitudinal regions for all months. Sampling errors are to be the largest for both refractivity and temperature parameters in the polar region due to fewer data samples and the uncertainties of ERA data. In addition, the number of RO events in the polar region is also relatively lower. In tropical regions, sampling errors are relatively low despite the decrease in the number of FY-3C events, which could be related to lower variability present in ERA data for the tropical region.

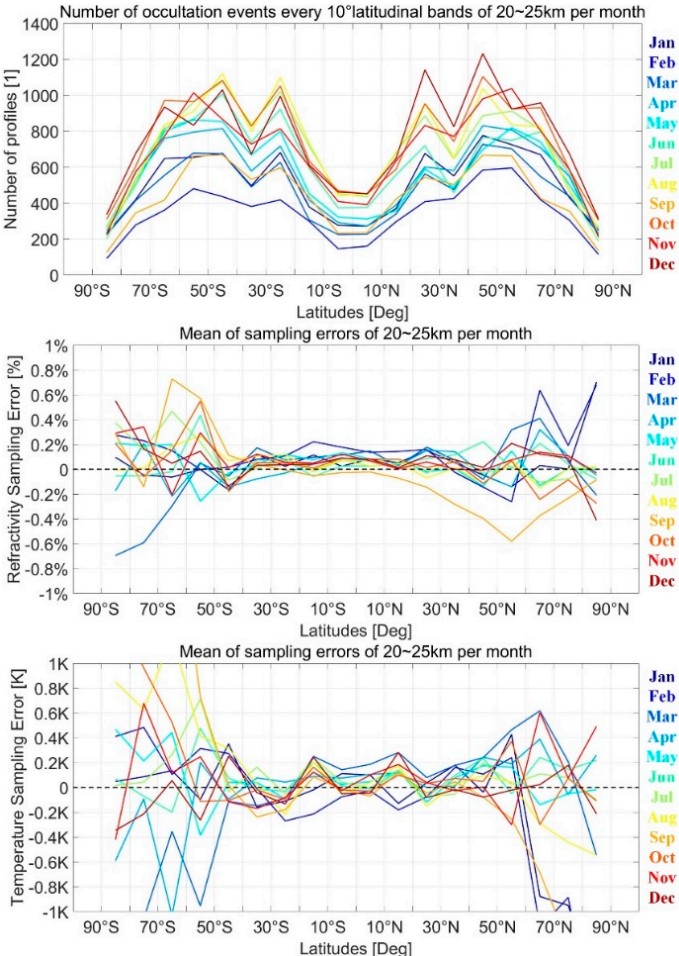

**Figure 10.** Monthly variation of occultation numbers in 20–25 km with 10° latitudinal bands in 2017 (upper panel), zonally averaged refractivity sampling error of altitude levels for 20–25 km altitude in 2017 (middle panel), zonally averaged temperature sampling error of altitude levels for 20–25 km altitude in 2017 (bottom panel).

## 4. Discussion

FY-3C RO profiles show an overall good performance compared to ERA-Interim data over the four year's period from 2015 to 2018. We found positive refractivity biases about 0.8 from 2 km to 5 km, and the origin of this biases require further study. We also found negative biases in refractivity profile in northern hemisphere polar region and positive biases in southern hemisphere polar region. This is probably related to the empirical observation uncertainties functions used by ROPP that are not fully suitable for the characteristics of FY-3C bending angles in polar region. Therefore, the retrieved optimized bending angles are biases and these biases are then propagated into refractivity [60,61].

Further efforts will be carried out to investigate an accurate estimation of uncertainties of FY-3C bending angles.

Furthermore, we also found temperature retrievals from FY-3C show consistent small negative biases for most cases. This is probably due to an inappropriate weighting of observed and background information in 1DVar process. The FY-3C data processing system uses ROPP to retrieve its 1DVar products. However, ROPP software estimates observation uncertainties based on empirical refractivity errors. However, these estimated observation uncertainties may not suitable for FY-3C products. Therefore, inappropriate weighting may result in systematic uncertainties.

## 5. Conclusions

In this study, FY-3C's RO profiles including refractivity, temperature and specific humidity for the period of 2015–2018 are evaluated by comparing RO profiles with collocated ERA-Interim profiles. Sampling errors of FY-3C refractivity and temperature profiles are also studied. Global retrieval errors for the years of 2017 and 2018 are smaller than that of 2015 and 2016. The systematic differences of global refractivity are generally within ±0.2% with the standard deviations are less than 2% in the 5–30 km altitude. The systematic differences of temperature show constant negative values varying from −0.2 to −0.4 K with standard deviations within 2 K in the altitude range of 5 to 20 km. Specific humidity systematic differences are within ±0.2 g/kg with corresponding standard deviations are overall less than 1 g/kg.

The latitudinal, seasonal, and altitudinal variations of FY-3C atmospheric profiles are also evaluated. It is found that both refractivity and temperature differences from ERA-Interim above 30 km are higher in winter season. Also, larger differences are found at higher altitudes and latitudes. Specific humidity differences are generally larger in tropical regions due to its higher content of water vapor. The sampling errors of FY-3C data are found to be higher at the higher altitudes and latitudes due to sparse data distribution in the polar region where the variability of the atmosphere could not be represented well.

In future, more efforts will be carried out to investigate the uncertainties of FY-3C bending angle, refractivity and temperature products, and also the uncertainties of background data. Based on the newly estimated uncertainties, it is expected that the high-altitude initialization and 1DVar processes will be updated to further improve the quality of FY-3C profiles.

**Author Contributions:** Conceptualization, J.W., Y.L. (Ying Li) and K.Z.; Data curation, Y.L. (Ying Li); Methodology, J.W. and Y.L. (Ying Li); Resources, Y.L. (Ying Li) and K.Z.; Software, J.W.; Supervision, Y.L. (Ying Li) and K.Z.; Validation, J.W., Y.L. (Ying Li), K.Z., M.L., W.B., C.L., Y.L. (Yan Liu) and X.W.; Visualization, J.W.; Writing—original draft, J.W. and Y.L. (Ying Li); Writing—review & editing, Y.L. (Ying Li) and K.Z. All authors have read and agreed to the published version of the manuscript.

**Funding:** At the CUMT: this research was funded by the National Natural Sciences Foundation of China (NSFC) (Grant No. 41874040), Xuzhou Key Project (ID: KC19111), Jiangsu dual creative talents and Jiangsu dual creative teams program projects of Jiangsu Province. At the APM/CAS, this research was funded by the Strategic Priority Research (SPR) Program of Chinese Academy of Sciences (Grant No. XDA17010304), NSFC (Grant No. 41604033), the National Key Research Program of China for "International Cooperation in Science and Technology Innovation" (No.2017YFE0131400). At NSSC/CAS, this research was funded by NSFC (41775034, 41405039), the Youth Innovation Promotion Association of the Chinese Academy of Sciences (Grant No. 2019151), the SPR Program of CAS (Grant No. XDA15021002) and the FengYun-3 (FY-3) Global Navigation Satellite System Occultation Sounder (GNOS) development and manufacture project.

**Acknowledgments:** We acknowledge the National Satellite Meteorological Centre, Chinese Meteorological Administration for providing FY-3C GNOS data, the ECMWF (Reading, UK) for providing access to their analysis and forecast data, and the RO software development and data processing teams of the RO data from their center used in the study.

**Conflicts of Interest:** The authors declare no conflict of interest.

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
