# Peer review of "An Evaluation of Fengyun-3C Radio Occultation Atmospheric Profiles Over 2015–2018"

_remotesensing, doi:10.3390/rs12132116_

Round 1

Reviewer 1 Report

See the attached PDF.

Reviewer 2 Report

Review of “An evaluation of Fengyun-3C radio occultation moist profiles over 2015-2018” by Wei et al.

It is good to see a paper evaluating FY-3C radio occultation (RO) data by comparing with the ERA-Interim reanalysis data set. However, there are a number of issues with this paper that make it unacceptable for publication in its present form.  If these are satisfactorily addressed, the paper would be a useful contribution to the literature and should be published.

General issues are:

  1. The English is not up to the standards of a scientific journal. I suggest that the authors get help from a person who is more fluent in English and edit it carefully. I realize this is difficult for authors for whom English is not their native language. I have made a number of suggested edits of the pdf file to help get started, but this is not a complete edit and the authors should check the entire paper carefully.
  2. This paper does not mention the well-known negative refractivity (N) bias in the lowest 1-2 km or so of the troposphere that is prevalent in the tropics due to super-refraction (ducting). This was first reported by Rocken et al. (1997) in the GPS-MET mission and has been found in every RO mission since then (e.g. Sokolovskiy, 2003; Ao et al. 2003; Xie et al., 2006; Xie et al., 2010; Xie et al., 2012; Anthes, 2011, and many other papers). Values are typically around -0.5% to -1% over large regions, with values exceeding -5% in regions were ducting (super-refraction) is prevalent (such as off the west coast of the U.S. or South America). This paper does show a small negative N bias in Figs. 4, 5 and 6 that reach close to -1%, but it is difficult to distinguish the different profiles in these figure (which need to be improved—see detailed comments on figures below). But Fig. 7 shows only a hint of a very thin layer of negative N biases near the surface in middle latitudes. Near the Equator in Fig. 7, there is little evidence of the negative bias near the surface, but there is a fairly deep layer of pronounced positive N biases reaching +2%. A small positive N bias has been found in other missions between about 2 and 5 km, but not this large. This should also be discussed. Possible reasons for the small negative N biases (besides low resolution of the figures) are in the details of the 1DVar procedure, which may be weighting the first guess refractivity too high and thus correcting the negative N biases in the RO retrievals. Another possibility is that the QC process removes many of the profiles with super-refraction. The number of profiles decreases very rapidly near the surface (Fig. 4).
  3. Some additional information on the 1D-Var procedure should be provided. For example, in the 1DVAR, are the observed RO refractivity values adjusted at all based on the background? It would be interesting to provide one or two examples of the original FY-3C RO N profile, the background N profile, and the final adjusted N profile from the 1DVar in a region of known super-refraction (e.g. the data point in Fig. 2 off the west coast of South America at latitude about 20°S).
  4. The variation of number of occultations with season is unexpected, and has not been reported before. For example, Fig. 5 SH refractivity count profiles at 20 km vary from about 2.0 E4 in JJA to about 1.0 E4 in DJF. Why is there such a large seasonal difference in number of occultations?
  5. Why are the number of profiles for T and q differences from ERA-I different than the number of refractivity profiles in Figs. 4, 5 and 6? And why are the numbers very different near the surface (the T numbers are constant all the way to the surface)?

Detailed comments and some suggested edits to improve the English are provided in the attached edited pdf file. It is difficult to edit pdf files so that there are some formatting issues in this edited file, but I hope most of the suggestions are clear and useful.

Some of the figures are difficult to read and should be improved. The colors of the profiles do not agree with the colors of the labels and the red and pink colors are hard to distinguish. The figures are especially difficult to read in the lowest 5 km, so perhaps expand the vertical scale from 0 to 5 km in Figs. 4-9 to make it easier to see the variations in the profiles below 5 km. This would extend the vertical size of the figures and the change (break) in vertical resolution at 5 km would have to be indicated.

Fig. 4: Too small and/or resolution is too low so it is difficult to distinguish profiles. Also color of 2015 profile looks much darker blue than the color of the 2015 label. The pink and red colors are difficult to distinguish. Perhaps just show three years to make it easier to see the different profiles (2015, 2016 and 2018).

Fig. 5: Resolution is too low and profiles are difficult to distinguish, especially below 5 km. The red and pink colors are too similar and the DJF profile looks very dark blue but the color is light blue-green.

Figs. 7 and 9: It is very difficult to see what is going on below 5 km.

Additional references:

Anthes, R. A., 2011: Exploring Earth's atmosphere with radio occultation: contributions to weather, climate and space weather, Atmos. Meas. Tech., 4, 1077-1103, doi:10.5194/amt-4-1077-2011.

Ao, C. O., T. K. Meehan, G. A. Hajj, A. J. Mannucci, and G. Beyerle (2003), Lower troposphere refractivity bias in GPS occultation retrievals, J. Geophys. Res., 108(D18), 4577, doi:10.1029/2002JD003216.

Dee, D. P., Uppala, S. M., Simmons, A. J., Berrisford, P., Poli, P., Kobayashi, S., Andrae, U., Balmaseda, M. A., Balsamo, G., Bauer, P., Bechtold, P., Beljaars, A. C. M., van de Berg, L., Bidlot, J., Bormann, N., Delsol, C., Dragani, R., Fuentes, M., Geer, A. J., Haimberger, L., Healy, S. B., Hersbach, H., Hólm, E. V., Isaksen, L., Kållberg, P., Köhler, M., Matricardi, M., McNally,

  1. P., Monge-Sanz, B. M., Morcrette, J.-J., Park, B.-K., Peubey, C., de Rosnay, P., Tavolato, C., Tépaut, J.-N., and Vitart, F.: The ERA-Interim reanalysis: configuration and performance of the

data assimilation system, Q. J. Roy. Meteorol. Soc., 137, 553– 597, doi:10.1002/qj.828, 2011.

Sokolovskiy, S. (2003), Effect of superrefraction on inversions of radio occultation signals in the lower troposphere, Radio Sci.38, 1058, doi:10.1029/2002RS002728.

Xie, F., S. Syndergaard, E. R. Kursinski, and B. M. Herman (2006) An approach for retrieving marine boundary layer refractivity from GPS radio occultation data in the presence of super-refraction. J. Atmos. Oceanic Technol., 23, 1629–1644, doi:10.1175/JTECH1996.1.

Xie, F., D. L. Wu, C. O. Ao, E. R. Kursinski, A. Mannucci and S. Syndergaard (2010) Super-refraction effects on GPS radio occultation refractivity in marine boundary layers, Geophys. Res. Lett., 37, L11805, doi:10.1029/2010GL043299.

Reviewer 3 Report

The main topic of this manuscript is on an evaluation of Fengyun-3C radio occultation moist profiles over the time period 2015-2018. This research evaluates the quality of the FY-3C RO products including refractivity, temperature, and specific humidity by comparing with the reanalysis data from the European Centre for Medium-Range Weather Forecasts (ECMWF). The topic is highly relevant and it is technically correct. The manuscript is very well written and organized. It deserves to published after minor corrections:
• Could you add a photo of Fengyun-3C satellite in the introduction?
• Could you add more details about the GNOS instrument?
• Improve the quality of all the figures.
• What is the improvement as compared to previous missions? This information should be clearly visible in the manuscript. Some tables?
• In the conclusions you say that: “The performance of FY-3C RO profiles is improved by time due to a stable working of hardware improved.” Could you add some comments within the text about hardware related topics? what are future improvements?
• In conclusions: “It is found that both refractivity and temperature errors are relatively higher in winter.” Why they are higher in winter? this point should be further discussed within the text.
• Do you think that the use of GNSS-RO and GNSS-R (e.g. CYGNSS) data could benefit the study of tropical cyclones? how?
I would like to congratulate the authors for this nice work.

Round 2

Reviewer 2 Report

The authors have made a good attempt to address the comments and suggestions from my previous review. The figures are improved and a discussion on the negative N bias has been added. I am sorry they did not receive my edited comments on the pdf file. I am not sure what happened. This time I am sending a Word file with my suggested edits directly to the corresponding author Kefei Zhang. I do not need to be anonymous.

The English is improved over the first version. However, I think there are a number of minor edits that would improve the text and some unnecessary sentences and phrases that can be deleted without loss of content. I have indicated some of these on my edited Word file.

There are some interesting results presented here and I think the paper is acceptable for publication after consideration of the following specific comments.

Detailed comments:

  1. Why did you remove ERA-Interim (ERA-I) from the abstract? And it should also be mentioned in Line 169. Also the title of Section 2.2—it was fine in the original version-why was it changed? There are three ECMWF reanalyses, ERA40, ERA-Interim and ERA5, so you should not just say ECMWF reanalysis. Please use ERA-Interim throughout.
  2. Line 67-GPS/MET was the first RO mission to measure Earth’s atmosphere, but not the first RO satellite. There were others long before 1995 to measure the atmosphere of other planets, including Mars.
  3. Lines 215-216—you might say why data from June-August 2018 are missing. And also, why is the number of ARP, ATP and AMP products different?
  4. Line 245—the horizontal footprint of RO is approximately 200-300 km. This is the horizontal scale represented by a single RO measurement. The horizontal resolution is given by the average spacing of RO observations and so depends on the number of RO satellites and their orbits.
  5. Lines 427-439—this means that below 5 km the temperature and specific humidity are weighted more heavily toward the background and hence the comparison with ERA-Interim is more of a comparison of the background to ERA-I than RO.
  6. I am not sure why the two sections Seasonal and Latitudinal Variations in the first draft were combined into one long section. I thought the original version was easier to read.
  7. Likewise, I am not sure why the introduction and discussion of sampling errors was moved outside the section that presents the sampling error results. I thought it read better the original way.
  8. Lines 506-507—I think the sentence “Its refractivity, temperature….climate applications” is unnecessary. Let the reader decide if these products are “good” and can be used for these applications.
